*electronics*

*Communication*

# A Design Fiber Performance Monitoring Tool (FPMT) for Online Remote Fiber Line Performance Detection

**Ahmed Atef Ibrahim [1,*], Mohammed Mohammed Fouad [2] and Azhar Ahmed Hamdi [2]**

[1] Electronics and Communication Engineering Department, Higher Technological Institute, 10th of Ramadan City 44629, Egypt
[2] Electronics and Communication Engineering Department, Faculty of Engineering, Zagazig University, Zagazig 44519, Egypt
* Correspondence: egog12@gmail.com

**Abstract:** A new technique for fiber faults events detection and monitoring in optical communication network systems is proposed. The fiber performance monitoring tool is a new proposed technique designed to detect, locate, and estimate the fiber faults without interrupting the data flow with efficient costs and to improve the availability and reliability of optical networks as it detects fiber faults remotely in real time. Instead of the traditional old method, the new proposed FPMT uses an optical time domain reflectometer to detect multiple types of fiber failures, e.g., fiber breaks, fiber end face contamination, fiber end face burning, large insertion losses on the connector and interconnection, or mismatches between two different types of fiber cables. The proposed technique methodology to detect the fiber failures depends on analyzing the feedback of the reflected signal and the pattern shape of the reflected signal over network fiber lines, supports a higher range of distance testing and performance monitoring, and can be performed inside an optical network in real time and remotely by integrating with an OSC board. The proposed technique detects fiber faults with an average accuracy of measurement up to 99.8%, the maximum distance to detect fiber line faults is up to 150 km, and it can improve the system power budget with a minimal insertion loss of 0.4 dB. The superiority of the suggested technique over real networks was verified with success by the Huawei labs' infrastructure nodes in the simulation experiment results.

**Keywords:** real time remote fiber faults; dense wavelength-division multiplexing; optical time domain reflectometer; optical performance monitoring; optical fiber



**Citation:** Ibrahim, A.A.; Fouad, M.M.; Hamdi, A.A. A Design Fiber Performance Monitoring Tool (FPMT) for Online Remote Fiber Line Performance Detection. *Electronics* **2022**, *11*, 3627. https://doi.org/10.3390/electronics11213627

Academic Editor: Mateusz Smietana

Received: 3 October 2022
Accepted: 2 November 2022
Published: 7 November 2022



## 1. Introduction

Data transmission lines via optical fiber cables are considered one of the most important methods used at the present time as they support the transmission of a large amount of data rates up to many terabytes per second over a long distance [1,2]. For this reason, it is necessary to find an easy and fast way to monitor performance and identify errors that occur in fiber cables without service interruption. In the dense wavelength division multiplexing system (DWDM), there are many fiber problems which effect the normal operation of the network (e.g., fiber breaks, fiber burning, curling, large bending, high attenuation, bit errors, dispersion, four wave mixing, absorption, splice attenuation, optical abnormality, fiber end face burning, etc.) [3,4]. In terms of the optical performance monitoring, the current solutions used to monitor and detect fiber line faults include the use of an optical time domain reflectometer (OTDR) [5,6]. This traditional OTDR method interrupts data traffic flow during the detection period and extends the time it takes for the network system to be restored and it is not financially viable due to the high maintenance and the operation expenses of an OTDR; including them in the system will significantly boost the system's cost [7,8]. The traditional method depends on offline detection mechanisms and an expert team is required on-site [9]. So, there is a great need for a fast online solution to check and monitor the fiber line faults remotely without interrupting the traffic

flow with efficient costs to allow the optical transmission network to be optimized and to reduce downtime [6–12]. In this paper, a new proposed technique for online remote fiber line performance is presented. The proposed technique is named as a fiber performance monitoring tool (FPMT). It is designed to remotely detect, locate, and estimate fiber line impairments with efficient costs, without interrupting data flow and with no need for an expert team on-site. The proposed technique was applied, and it detected multiple types of fiber faults such as fiber breaks, fiber end face contamination, fiber end face burning, and large insertion losses on the connector and the reflection peak due to interconnection or mismatches between two different types of fiber cables.

### *1.1. Related Research and Literature Review*

Returning to previous research, we found that the process of fiber line performance monitoring (FLPM) (optical performance monitoring (OPM)) can be divided into two categories: offline performance monitoring and real time performance monitoring (RTPM) (online performance monitoring).

Vivak et al. [11] presented an algorithm that uses the total loss measured by OTDR to predict the actual position of the defect instead of measurement of the pulse fraction of a probe made of silica fiber that is scattered back by Rayleigh scattering. Due to the very low levels of backscatter in a single mode fiber (SMF) at long wavelengths, very sensitive optical detection is required to obtain an adequate performance.

The traditional method, OTDR, is the most common method used in the optical network and for the localization of the OTDR failure in the optical network and is performed manually which requires more effort and time [12,13].

A. Bakar et al. [8] explained a new online technique called the fiber break monitoring system (FBMS) which was built to detect fiber cuts (break) online and fast with low costs and an acceptable location measurement accuracy.

In the studies in [14–17], the authors proposed a real time fault detection by monitoring the main signal parameters as optical signal-to-noise ratio (OSNR), chromatic dispersion (CD), mode coupling (MC), polarization mode dispersion (PMD), modulation formats (MF), and forward error correction (FEC) codes, then predicting the failures by any changes in measurements of the signal parameter values.

M. Amirabadi et al. [18] also proposed online performance monitoring by monitoring the physical layers in optical networks to analyze the data and the status of the layers, then provide the feedback to the controller to predict the failures.

The real time performance monitoring based on Fourier transform spectrum analysis (FTSA) was studied in [19]. This technique was designed for OSNR online monitoring.

Against that background, the framework suggested by the authors is based on a new technique for real time performance monitoring that detects, locates, and estimates all hard failures (HF) (fiber breaks, fiber end face contamination, fiber end face burning, large insertion loss on the connector, etc.) through the optical network.

The other studies used real time performance monitoring based on soft failure (SF) measurements (OSNR, MC, CD and etc.) to predict failures and alarms. Other studies used real time performance monitoring based on hard failure (HF) measurements such as fiber breaks only [8,20].

### *1.2. Aim of the Paper*

In this paper, a new proposed technique for online remote fiber line performance monitoring is presented to overcome the drawbacks of the traditional OTDR method and the limitations of new real-time technologies. The implemented technique methodology detects the fiber faults over fiber lines remotely and in real time based on an analysis of the feedback of the reflected signals and the shape (pattern) of the reflected signal for each fault. The reflected signals have specific criteria, and are selected by implementing the FPMT technique before transmission, and are called reflected test signals.

The proposed technique can be applied to detect, locate, and estimate all optical hard failures (HF) (fiber breaks, fiber end face contamination, fiber end face burning, large insertion loss on the connector, the reflection peak due to interconnection or mismatch between two different types of fiber cables, etc.) through a passive optical network (PON) and a single mode fiber (SMF). The proposed technique is defined by low costs, detects online and remotely without interrupting the data traffic flow, improves the system power budget with a low insertion loss, detects the fiber line over a long distance, detects fiber fault localization with a high accuracy of measurement, and detects multiple types of HF fiber faults.

### 1.3. Paper Organization

The remainder of this paper is structured in the following manner. In Section 2, the characteristics and parameters of the optical network system in which the FPMT technology applied are introduced. In Section 3, the working principle and signal flow over the proposed FPMT technique and the parameters of the proposed technique and the test signal generated are presented. Section 4 is dedicated to an illustration of the ways of transmitting a test signal on a fiber line and the parameters that affect it at the maximum distance of the test signal transmitted by using the OSC board to detect fiber line failures. The final results and discussion of several evaluation techniques are applied in Section 5. Finally, the final recommendations and remarks are provided in Section 6.

## 2. FPMT Technique

To apply the FPMT for monitoring and analyzing the fiber path and collecting the information by using ultra-high speed data transmission optical system, a DWDM system is implemented with high-capacity wavelength division multiplexed transmission (WDM) technologies [1,2]. Figure 1 shows the ultra-high speed data transmission optical system structure with the implementation of an 80-channel DWDM system able to transmit and receive up to terabytes per second over single mode optical fiber pair cables.

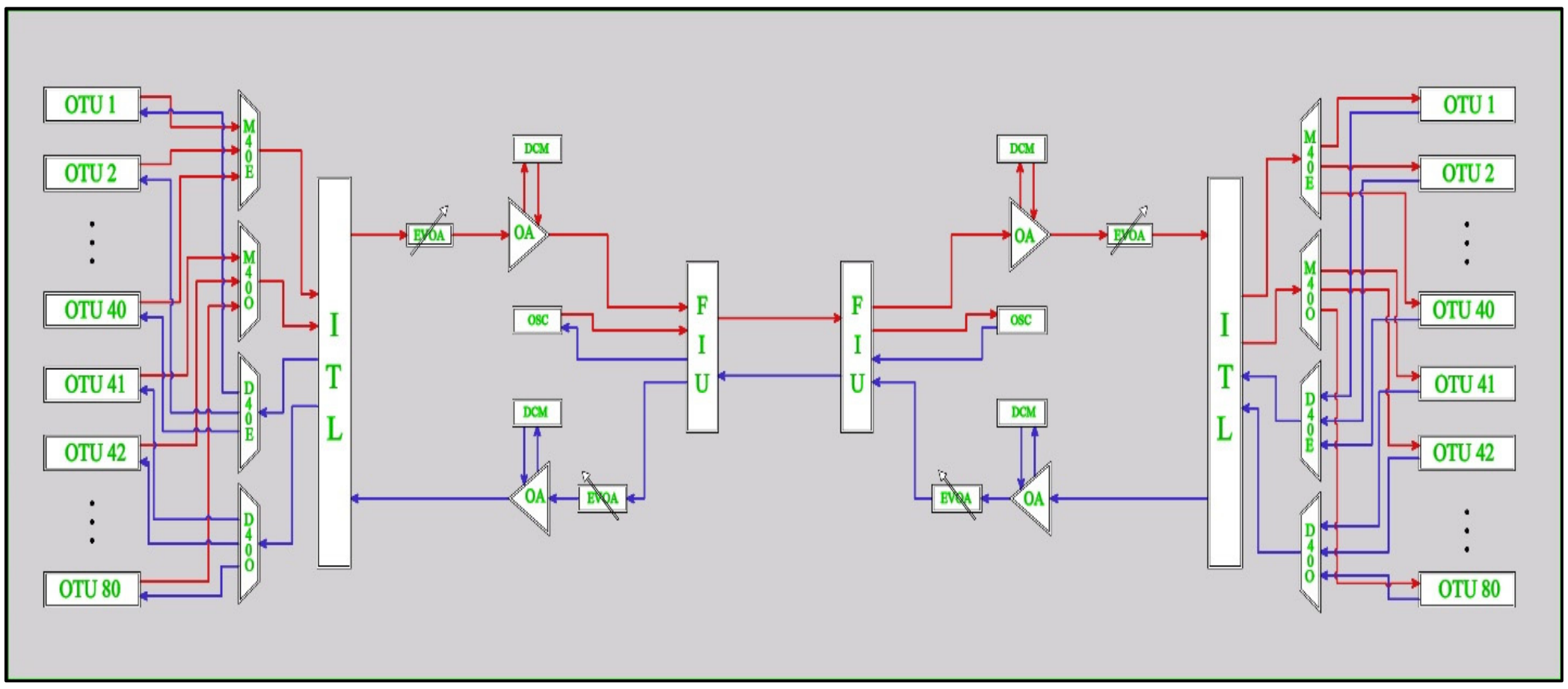


**Figure 1.** The ultra-high speed data transmission optical system structure by implementing an 80-channel DWDM system.

The implemented DWDM system consists of several main constituents. The optical transponder unit (OTU) is used to receive the customer data and convert these data to the DWDM frequency band (C-band) standardized by international telecommunication union (ITU) and sends them to the unit of multiplexer units (MU). There are two types of MUs used: the even and odd ones. Each multiplexer can send and receive up to 40 colored channels, so we have 80-channels (40 even and 40 odd). The M40 is an example of the multiplexer unit. D40 performs the inverse process by receiving the direction through de-multiplexing the 40-channels and sends them separately to the OUTs. The interleaver

board (ITL) is used to multiplex/combine the even and odd frequency channels in the transmit side path and de-multiplex/distribute them in the receiving side. The electrical variable optical attenuator (EVOA) is used to control the input power before reaching the optical amplifier by adding an attenuation value based on the current situation and the application scenario and its remote network management system (NMS) are controlled. The optical amplifier (OA) is used to boost the signal power and allow it to travel over a long distance. The Raman amplifier or erbium-doped fiber (EDFA) amplifier can be used. The dispersion compensation module (DCM) is used to compensate the dispersion of the signal to enhance the system quality and performance. The optical supervisory channel board (OSC) is used to add/terminate the management signals, especially the optical overhead bytes and information such as the optical channel (OCh), optical multiplex section (OMS), and optical transmission section (OTS) overheads. The OSC board works outside the C band and the OSC signals cannot be amplified; instead, it sends and receives using sufficient power and a receiver sensitivity of 1491 nm and 1510 nm. The fiber interface board (FIU) is used to combine/multiplex the traffic signal and OSC signal and couple them in a single line fiber and de-multiplex them when received into two paths. It is suggested that the FPMT is connected for monitoring and analyzing the fiber path and collecting the information through a DWDM system shown in Figure 2.

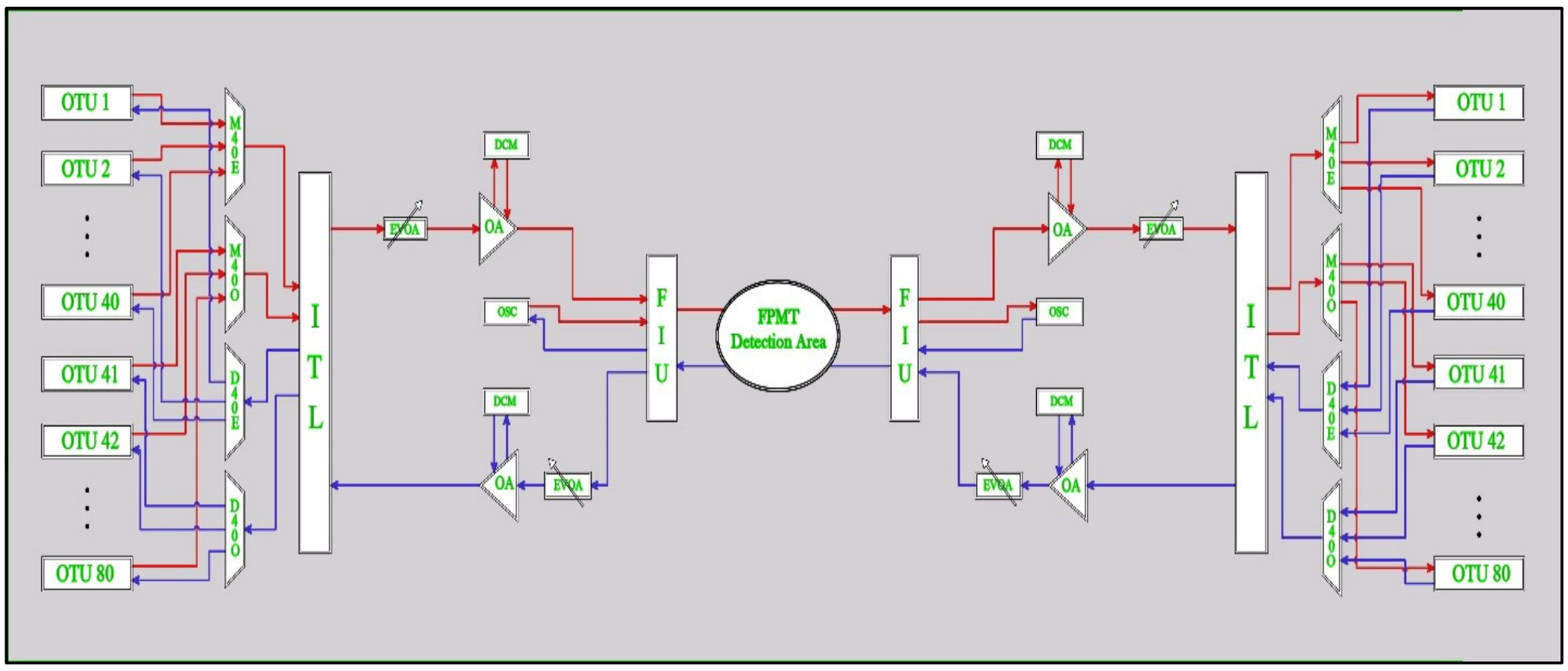


**Figure 2.** FPMT detection area over the implemented DWDM system.

The specifications for the implemented DWDM system for monitoring and analyzing the fiber line are illustrated in Table 1.

**Table 1.** The specifications of the implemented DWDM system.

| Parameter | Specification |
|---|---|
| Cabinet name | OSN |
| Model | 9800 |
| Type of Subrack | universal platform |
| Number of wavelengths | up to 80 channels |
| Data rate | up to 100 Gbps/channel |
| Attenuator | VOA adjust power remotely |
| Optical amplifier | OBU in TX–OAU in Rx |
| OSC board | ST1 |
| Multiplexer | 2 × M40 |
| De-multiplexer | 2 × D40 |
| Transponder | LDX |
| Dispersion module | DCF |

## 3. Working Principle of the FPMT Technique

The FPMT detection circuit is implemented to generate the test signal and transmit it with the traffic signal flow to detect any fiber faults through the fiber line online and remotely. The FPMT circuit as shown in Figure 3 consists of several main constituents. The random sequence bit generator (RSG) generates a predefined digital stream. The laser diode (LD) receives a digital stream from RSG and transmits the FPMT test signals through the same fiber line by using another frequency band to avoid any interference with the traffic signal transmitting on the C band. The hybrid filter (HF) distinguishes between the transmit and receive path signals that filter the receiving reflected signal. The photo detector (PD) receives the filtered reflected signals and converts them to digital streams then to the digital signal processing module to analyze.

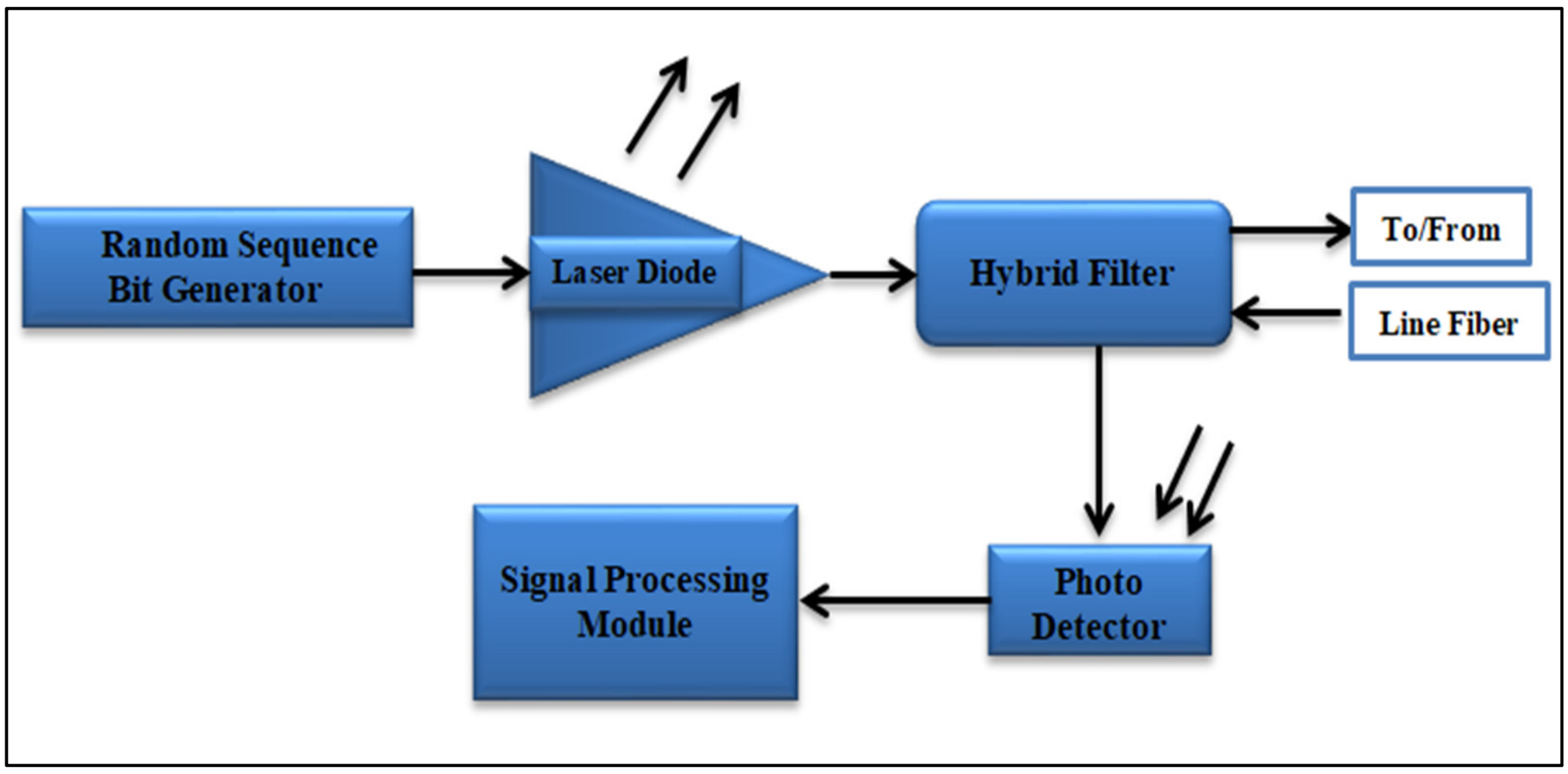


**Figure 3.** Proposed FPMT detection circuit block daigram.

We can monitor and analyze signal feedback based on the power level, receiving time, bit error, shape of the reflected signal, and all other signal parameters used in the detection, and then we can define the fiber line status and characteristics.

The implemented FPMT technique parameters for monitoring and analyzing the fiber line are illustrated in Table 2.

**Table 2.** The parameters of implemented FPMT technique.

| Parameter | Value |
|---|---|
| Transmit power | −3 to −5 dBm |
| Receiver sensitivity | −43 dBm |
| Insertion loss | −0.4 dB |
| Time slot | 32 Time slot/frame |
| Capacity of time slot | 8 bit |
| Stream of bit | 2 Mbps |

Figure 4 clarifies the steps of the working principle and signal flow over the proposed FPMT technique that can be summarized as follows: the DWDM system starts by sending the traffic that is coming from the customer side (voice, video, data) over the C-band wavelength (1525–1565 nm). The traffic will mix/multiplex with the test signal generated by the FPMT system that is integrated with the OSC board and with the control/management signal generated by the OSC board too. All these signals are on a different wavelength range to avoid interference (the OSC signal wavelength is at 1510 nm), then are transmitted over the line fiber and received on the other side. Any reflected signals caused by fiber cuts,

fiber burns, fiber contamination, etc., are reflected back to the transmitter side and filtered from the traffic signal by the FPMT technique and then analyzed to confirm the root cause.

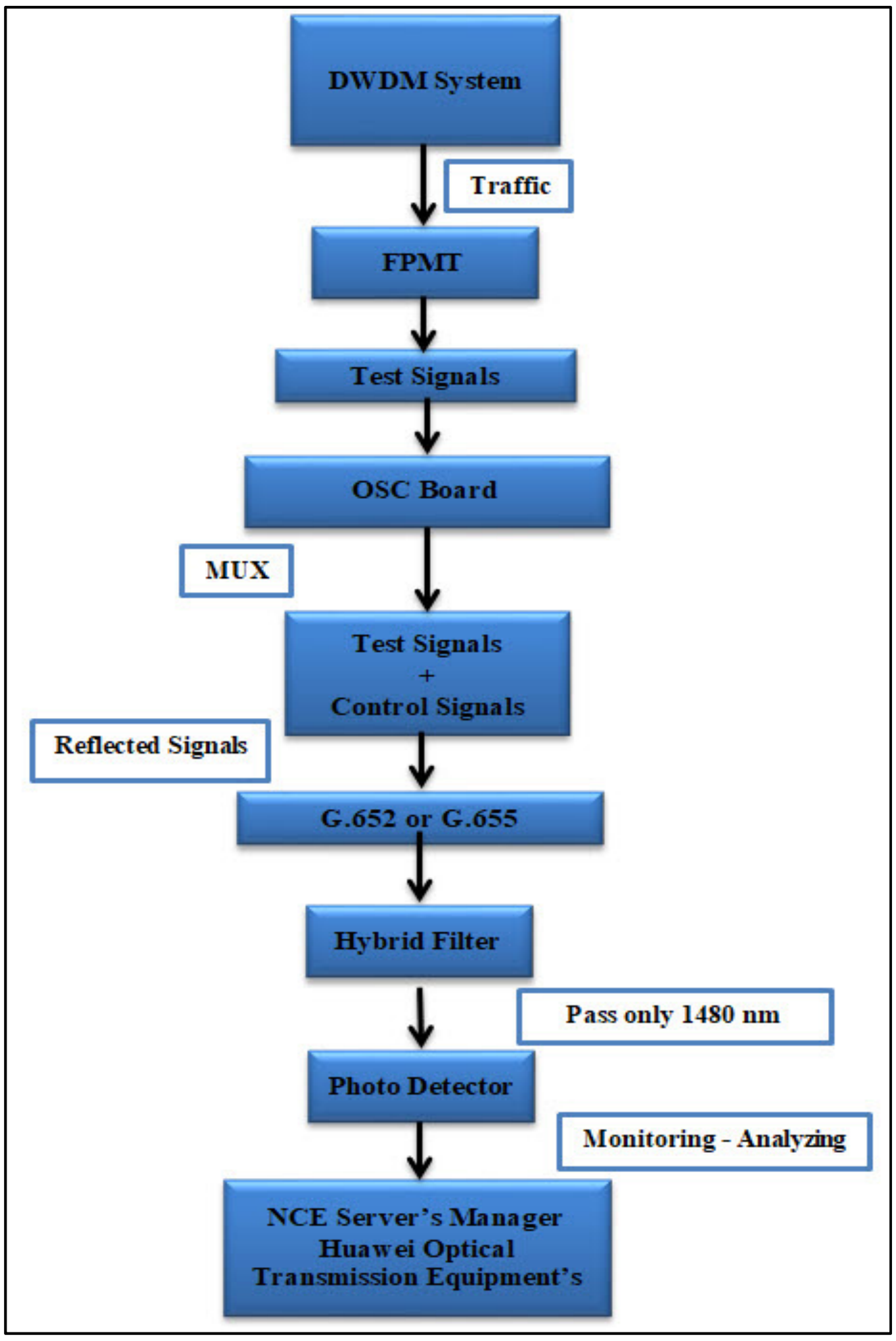


**Figure 4.** The working principle and signal flow over the proposed FPMT technique.

## 4. The Method to Transmit Test Signals

The test signals produced by implementing the FPMT technique can be transmitted over the OSC signal in the optical DWDM network in three ways as follows.

The first way involves carrying the FPMT signal at a different wavelength range from the OSC frame to ensure that there is no interference with the traffic signal wavelength ranges of the C band from 1525 nm to 1565 nm and the OSC signal wavelength ranges of 1510 nm or 1490 nm. Then the FPMT signal is integrated with the OSC signal and the traffic signal over the same line fiber.

The second way involves transmitting the FPMT signal at the same wavelength range of OSC signal as 1490 nm or 1510 nm inside specific time slots in the OSC frame that is idle and does not carry any data. This method will save the wavelength range and also utilize the frequency usage.

The third way involves sending the test signals at any idle time slots of the OSC frame and uses a different wavelength range of the OSC frame and wavelength ranges of C band to avoid interference with control and traffic signals. This method can be considered a mixture between the first method and second method and is safer for any type of interference. In this research, we used the first way to obtain the fast and simple detection of the fiber line and the test signal wavelength at 1480 nm was used to avoid any interference during the applied FPMT-implemented circuit.

*4.1. The OSC Board before and after Integrating the Implemented FPMT Technique Detection Circuit*

The OSC board block diagram before integrating the FPMT detection circuit to the design is shown in Figure 5.

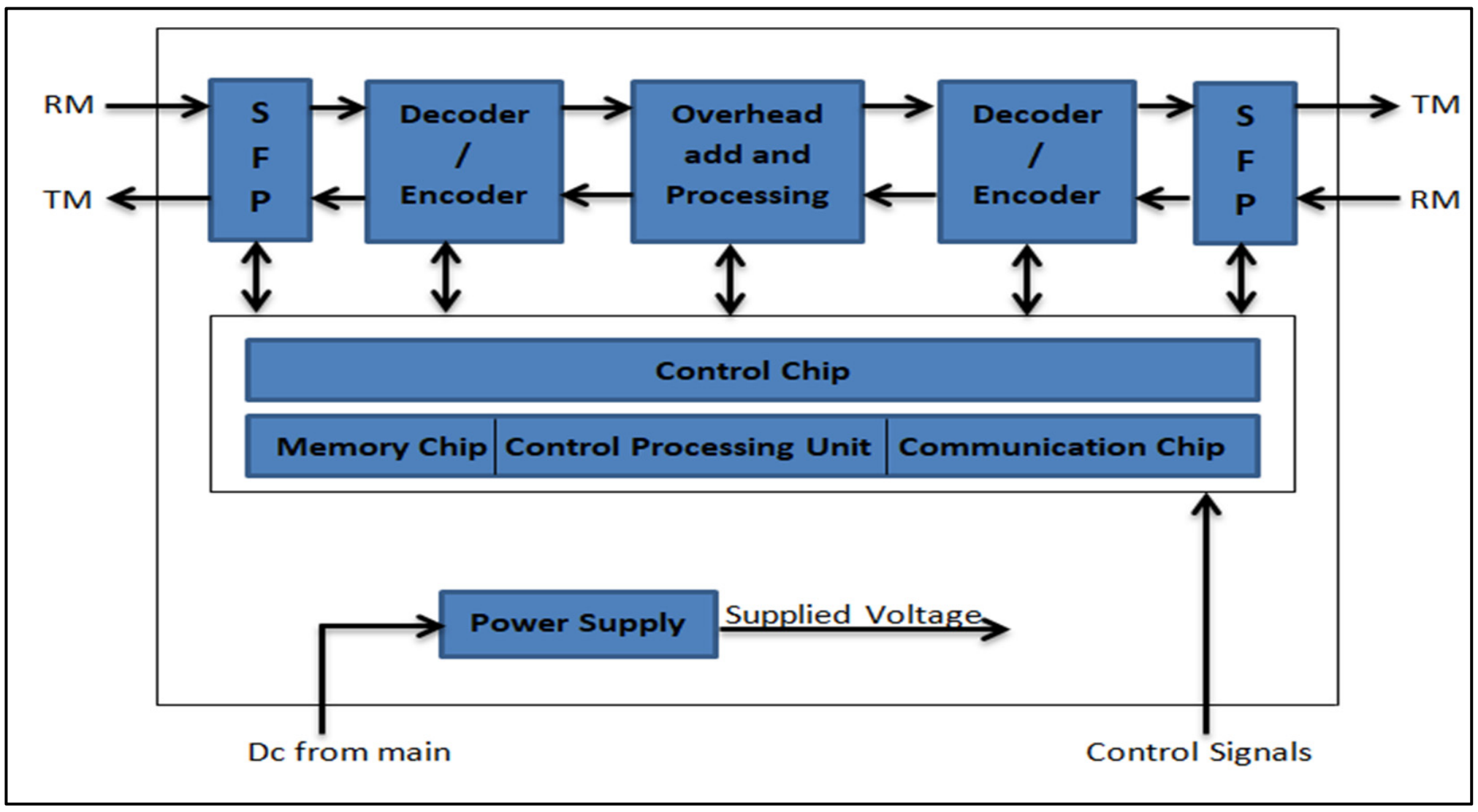


**Figure 5.** The OSC board block diagram before integrating the FPMT detection circuit.

The OSC frame is a 32-time slot in 125 microseconds with a 2 Mbps rate and the wavelength window 1490 nm and 1510 nm is used [21–23]. The small form factor pluggable (SFP) is the optical module used to convert the optical signals into the electrical signals and vice versa. The TM and RM are the transmit management signal ports and receive management signal ports, respectively [21,23].

The OSC board is an active element that has no insertion loss but when the optical supervisory channel (OSC) is used the extra power of the fiber line units, which is considered to be 1 dB (the insertion loss of the fiber interface boards (FIUs), at the two ends) should be considered in the budget [22].

To achieve the transmission of the test signal generated from the FPMT technique over the OSC signal on the same fiber line in the optical DWDM network, the implemented FPMT detection circuit should be integrated with the OSC board, as shown in Figure 6.

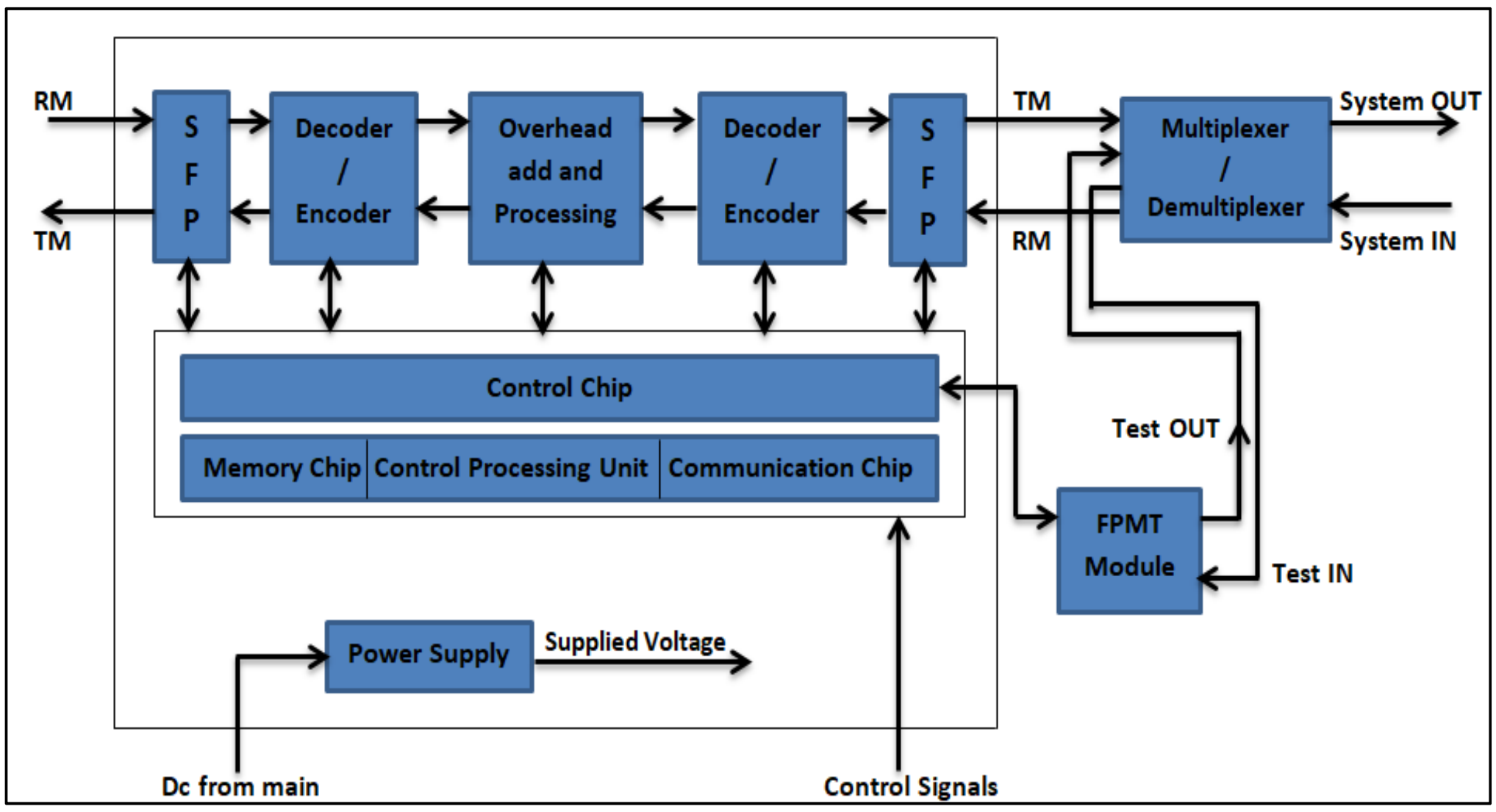


**Figure 6.** The OSC board block diagram after integrating the FPMT detection circuit.

The OSC board specifications and parameters are shown in Table 3.

**Table 3.** The specifications and parameters of OSC board.

| Parameter | Specification/Value |
|---|---|
| Cabinet | OSN 9800 |
| OSC board | ST 1 |
| OSC frame | 32 Time Slot |
| Wavelength | 1490 nm or 1510 nm |
| Insertion loss | −1.00 dB |
| Receiver sensitivity | up to −48 dBm [21,22] |
| Maximum transmit power | −7 dBm [21,22] |
| Maximum attenuation loss | −34 dB derived in Section 4.2 |

*4.2. Maximum Distance of the Test Signals Transmitted by the OSC Board*

The multiple real optical transmission networks were tested and these networks were monitored based on the performance results and the parameter design at Net-Star WDM for Huawei and APT for Nokia.

We derived the equation with the mentioned loss types and parameters that affect an optical signal crossing the network and the maximum distance to transmit the signal.

$$P_{Rx} = P_{Tx} + \alpha_{FL} \quad (1)$$

$$\alpha_{FL} = \alpha_{Max} + EFM_{Max} + \alpha_{B} \quad (2)$$

where $P_{Rx}$ represents the minimum received power sensitivity, $P_{Tx}$ represents the transmitted power, $\alpha_{FL}$ represents the attenuation at the fiber line, $\alpha_{Max}$ represents the maximum attenuation-based distance, $EFM_{Max}$ represents the maximum effective system fade margin, and $\alpha_{B}$ represents the other basic attenuations over the fiber line.

The maximum effective system fade margin can be considered as −4 dB and the maximum other basic attenuation over the fiber line as (bending, scattering, nonlinear effects, others) can be considered as −3 dB.

The maximum transmitted output power from the OSC board is −7 dBm and the minimum received power sensitivity at the OSC board is −48 dBm [21,22].

In this case we used the OSC board, and the results were as follows:

When substituted into Equation (1), the attenuation at the fiber line was −41 dB.

When substituted into Equation (2) the maximum attenuation loss-based distance was −34 dB.

$$\alpha_{Max} = D_{Max} \times \alpha_{p} \quad (3)$$

where $D_{Max}$ represents the maximum distance to transmit signals and $\alpha_{p}$ represents the fiber line attenuation coefficient.

The test signal over fiber lines was carried by a single mode fiber (SMF), G.652 or G.655, as explained in Section 5.

The better wavelength performance during transmission at SMF G.652 or G.655 was 1550 nm, which gave the best attenuation coefficient at approximately −0.22 dB per kilometer [24,25].

The maximum distance to transmit a test signal was obtained by using the OSC board exposed to extra attenuation from the FPMT technique insertion loss and the FIU insertion loss. Both were connected to the OSC board. Equation (4) shows all these attenuation losses.

$$\alpha_{Max} = (D_{Max} \times \alpha_{p}) + \alpha_{FIU} + \alpha_{FPMT} \quad (4)$$

where $\alpha_{FIU}$ represents the FIU insertion loss at about −1 dB [22] and $\alpha_{FPMT}$ represents the FPMT technique insertion loss at about −0.4 dB.

In this case we used the OSC board to transmit a test signal, substituted it into Equation (4), and the result was as follows: the maximum distance to transmit a test signal by using the OSC board for detecting fiber line failures was approximately 150 km.

From Equations (1)–(3), there were three important parameters that affected the maximum distance of the test signal transmitted by using the OSC board to detect fiber line failures including the OSC board receiver sensitivity (minimum receive power), the OSC board maximum transmit power, and the fiber attenuation coefficient.

Figure 7 shows the maximum distance to transmit the signal between two nodes and the signal flows over the fiber line.

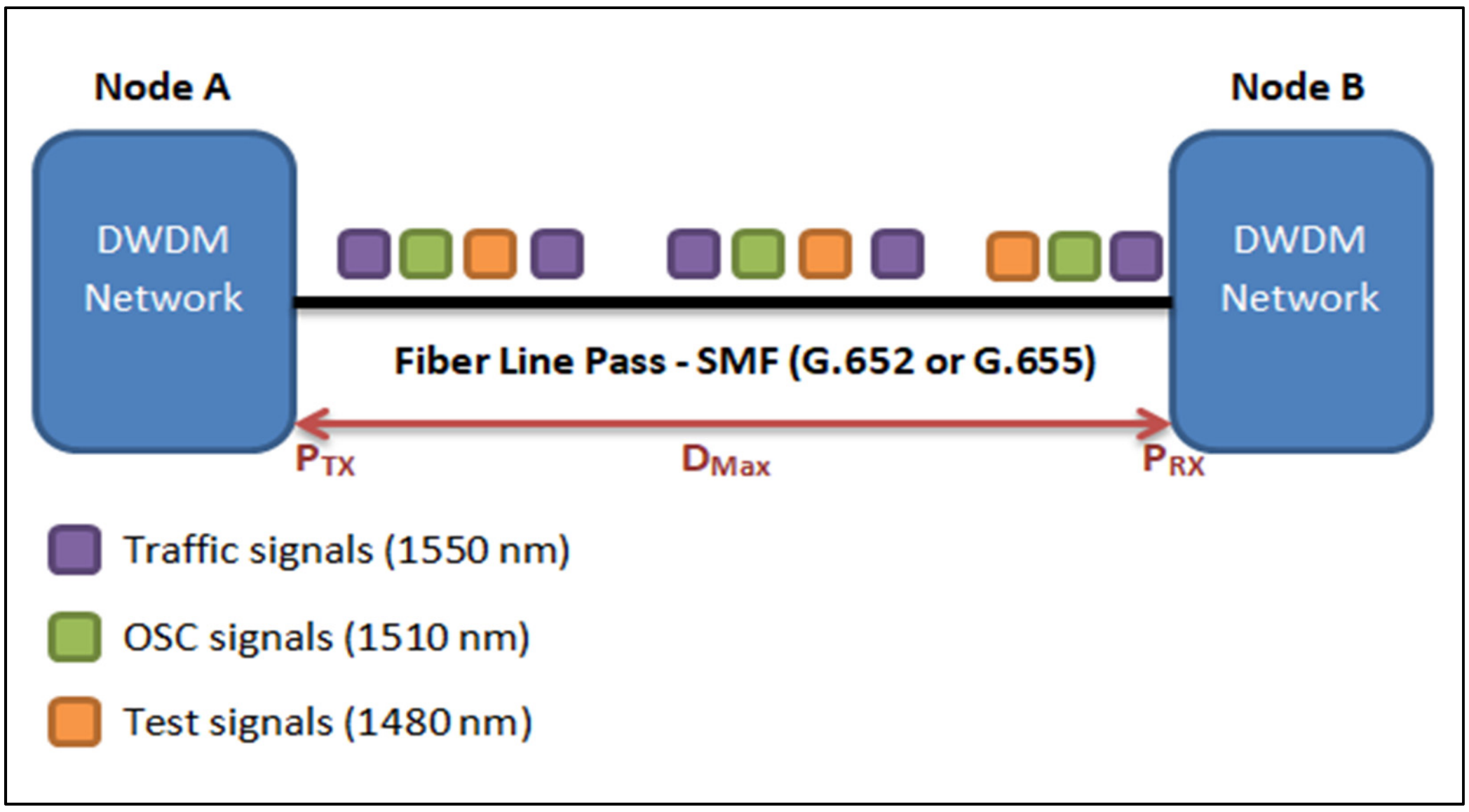


**Figure 7.** All the signal flows over the fiber line and the maximum distance to transmit the signal between two nodes.

## 5. Results and Discussion

The practical results collected by Huawei network cloud engine (NCE) server at the optical transmission equipment were according to the reflected test signals analyzed [26,27]. The system was implemented through remote access Huawei labs infrastructure nodes. The specification parameters of the test signals generated by the FPMT are shown in Table 4.

**Table 4.** The specifications of the test signals.

| Parameter | Specification |
|---|---|
| Applied by | DWDM system |
| Generate by | FPMT |
| Transmitted by | OSC board |
| Shape | Random Binary Bits |
| Pulse width | 20,000 Nanoseconds |
| Signal rate | 2 Mbps |
| Wavelength | 1480 nm |
| Fiber attenuation per Km | 0.22 to 0.35 dB |
| Fiber standard type | G.652 or G.655 |
| Maximum detected distance | 150 Km |

The specifications of the implemented system for monitoring and analyzing the fiber line and collecting the information are shown in Tables 1 and 5.

**Table 5.** The specifications of the implemented system for monitoring and analyzing the fiber line and collecting the practical results.

| Parameter | Specification |
|---|---|
| Applied by | Huawei Lab |
| Monitoring and analyzing fiber line data collection by | NCE server's Huawei |
| Standard types of optical fiber used | SMF G.652 or SMF G.655 |
| Wavelength used at (G.652 or G.655) | 1550 nm |
| Fiber attenuation coefficient per Km | Approx. 0.22 dB |
| Actual fiber fault types created on the applied DWDM system | (a) Fiber break<br>(b) Fiber end face contamination<br>(c) Fiber end face burning<br>(d) Large insertion loss on the connector<br>(e) Interconnection between two type fiber cable G.652 and G.653 |

The system used the standard types of optical fibers specified by ITU [24,25]. The practical results were applied to SMF whose fiber cable types were G.652 or G.655 with specific specifications, as shown in Table 6.

**Table 6.** The specifications of single mode fiber standard types specified by ITU.

| Type | G.652 | G.655 |
|---|---|---|
| Definition | Standard SMF<br>Zero dispersion point is about 1310 nm [28] | Non-zero dispersion shifted fiber [28].<br>Zero dispersion point is shifted away from 1510 nm |
| Wavelength | 1310 nm–1550 nm | 1310 nm–1550 nm |
| Maximum attenuation coefficient | Attenuation of 1310 nm ranges from 0.3 to 0.4 dB/Km and the attenuation of 1550 nm ranges from 0.17 to 0.25 dB/Km | The attenuation of 1550 nm ranges from 0.19 to 0.35 dB/Km |
| Application | Used in SDH and DWDM | Used in SDH and DWDM |

## *5.1. Practical Results*

The practical results detected five types of HF. These failures were already created on the applied system to check the possibility of the test signals detecting fiber line faults. The practical results after the proposed FPMT technique was applied were as follows.

($a_1$) Fiber breaks: It was already known that there was a fiber break created on the real applied system at a distance 5.2 km and a review of the practical results using the FPMT technique are shown in Figure 8. The reflection peak was about 5.22 km and the reflection value was −5 dB which indicated a specific pattern expressing the fiber break failure. The test was performed on the standard type of SMF-G.652.

($a_2$) Another case of a fiber break: It was already known that there was a fiber break created on a real applied system at a distance of 9 Km and the practical results using the FPMT technique are shown in Figure 9. The reflection peak was about 9 km and the reflection value as −20 dB which indicated a specific pattern expressing the fiber break failure. The test was performed on the standard type of G.655.

| | |
|---|---|
| Test signal transmitted by | OSC board |
| SMF Standard type is | SMF-G.652 |
| Wavelength used | 1550 nm |
| Failure detected by FPMT | Fiber break with a specific pattern drawn |
| FPMT fault location distance | Approx. 5.2 Km |
| Reflection value | Approx. −5 dB |

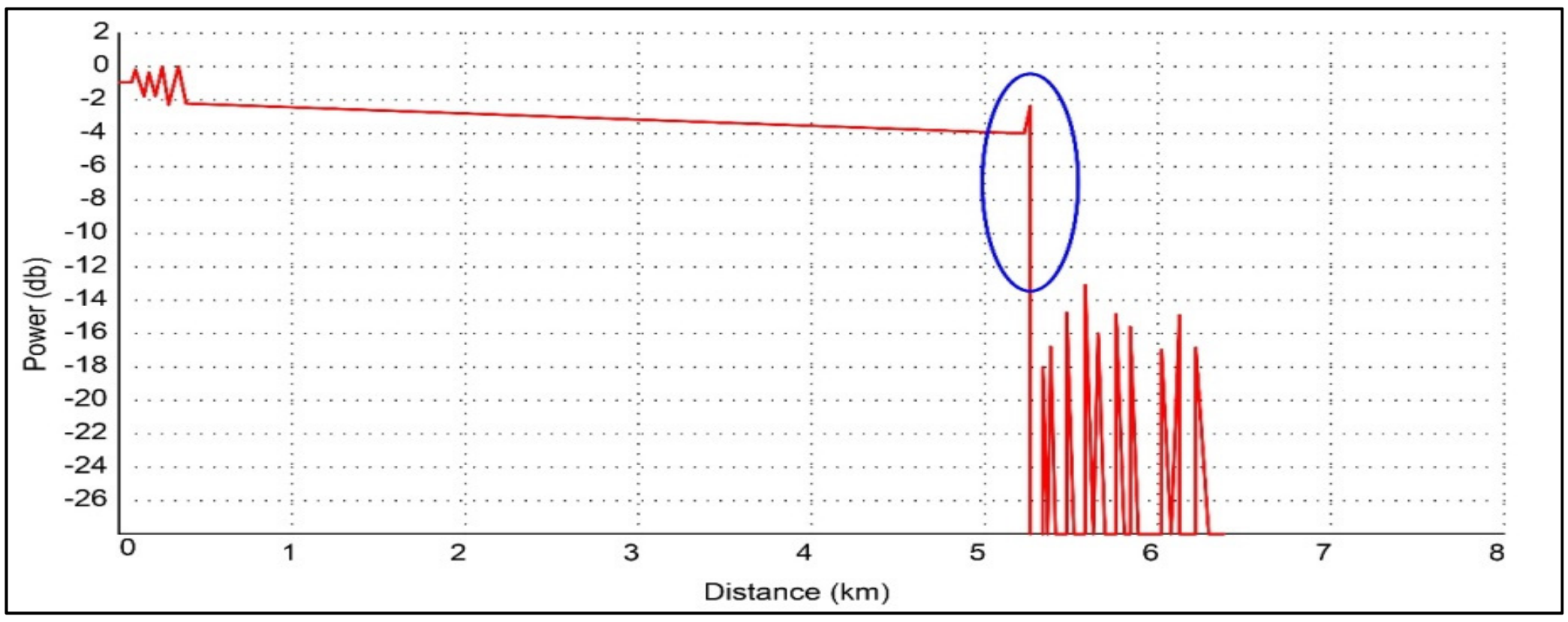


**Figure 8.** Fiber break event (standard SMF G.652 at 1550 nm used and FPMT test signal was transmitted by the OSC board).

| | |
|---|---|
| Test signal transmitted by | OSC board |
| SMF Standard type is | SMF-G.655 |
| Wavelength used | 1550 nm |
| Failure detected by FPMT | Fiber cut with a specific pattern drawn |
| FPMT fault location distance | Approx. 9 Km |
| Reflection value | Approx. −20 dB |

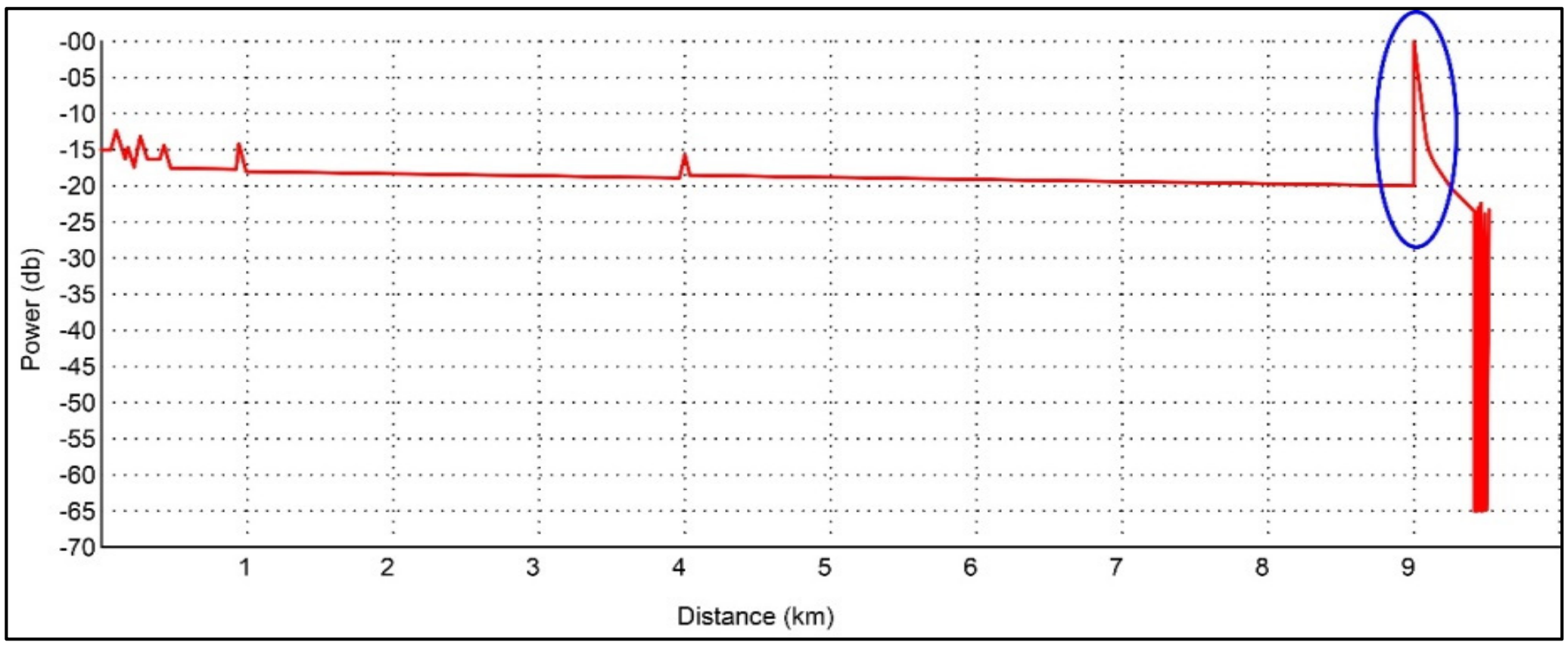


**Figure 9.** Fiber break event (standard SMF G.655 at 1550 nm was used and FPMT test signal was transmitted by the OSC board).

(b) Fiber end face contamination: It was already known that there was a fiber end face contamination created on a real applied system at a distance of 2 Km and the practical results using the FPMT technique are shown in Figure 10. The reflection peak was about 0.2 km and the reflection value was −7 dB which indicated a specific pattern expressing the fiber end face contamination failure. The test was performed on the standard type of SMF-G.652.

| | |
|---|---|
| Test signal transmitted by | OSC board |
| SMF Standard type is | SMF-G.652 |
| Wavelength used | 1550 nm |
| Failure detected by FPMT | Fiber end face contamination with specific pattern drawn |
| FPMT fault location distance | Approx. 0.2 Km |
| Reflection value | Approx. −7 dB |

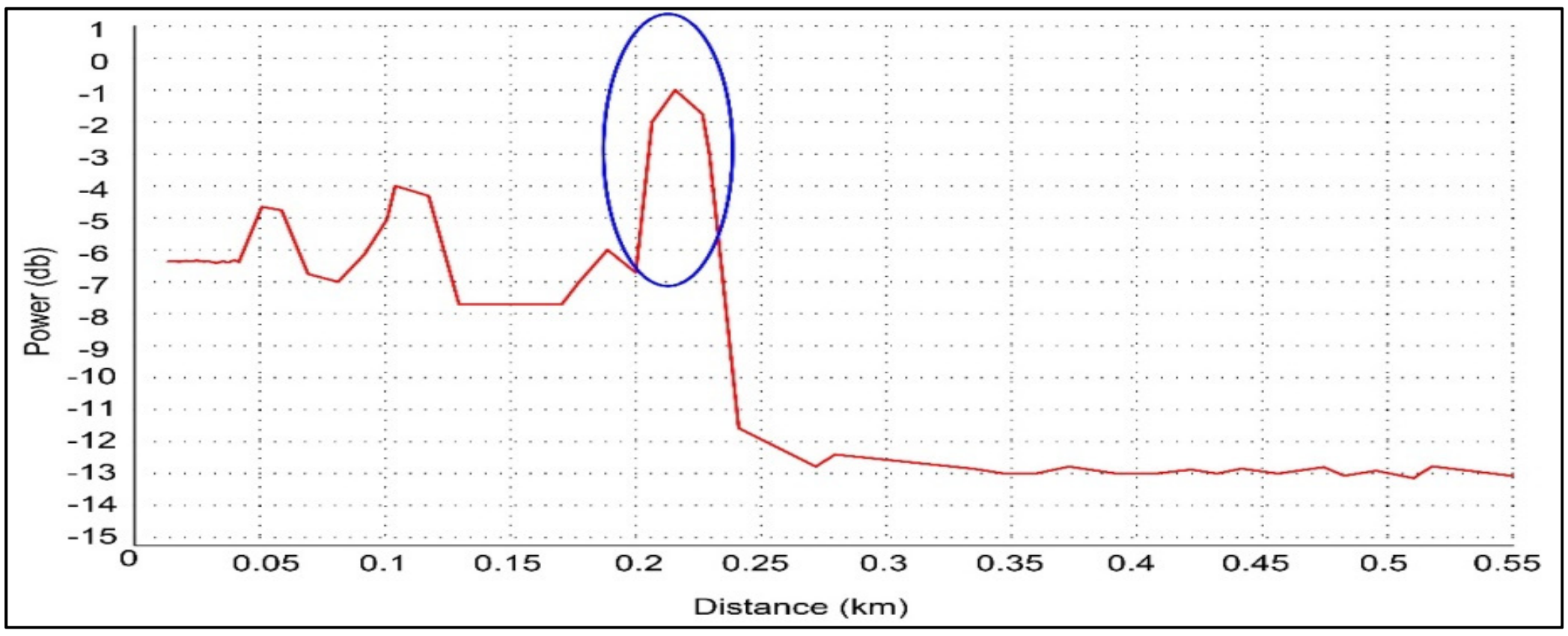


**Figure 10.** Event is Fiber end face contamination (standard SMF G.652 at 1550 nm was used and FPMT test signal was transmitted by the OSC board).

(c) Fiber end face burning: It was already known that there was a fiber end face burning created on a real applied system at a distance of 0.5 Km and the practical results using the FPMT technique are shown in Figure 11. The reflection peak was about 0.45 km and the reflection value was −14 dB which indicated a specific pattern expressing the fiber end face burning failure. The test was performed on the standard type of SMF-G.655.

| | |
|---|---|
| Test signal transmitted by | OSC board |
| SMF Standard type is | SMF-G.655 |
| Wavelength used | 1550 nm |
| Failure detected by FPMT | Fiber end face burning with a specific pattern drawn |
| FPMT fault location distance | Approx. 0.45 Km |
| Reflection value | Approx. −14 dB |

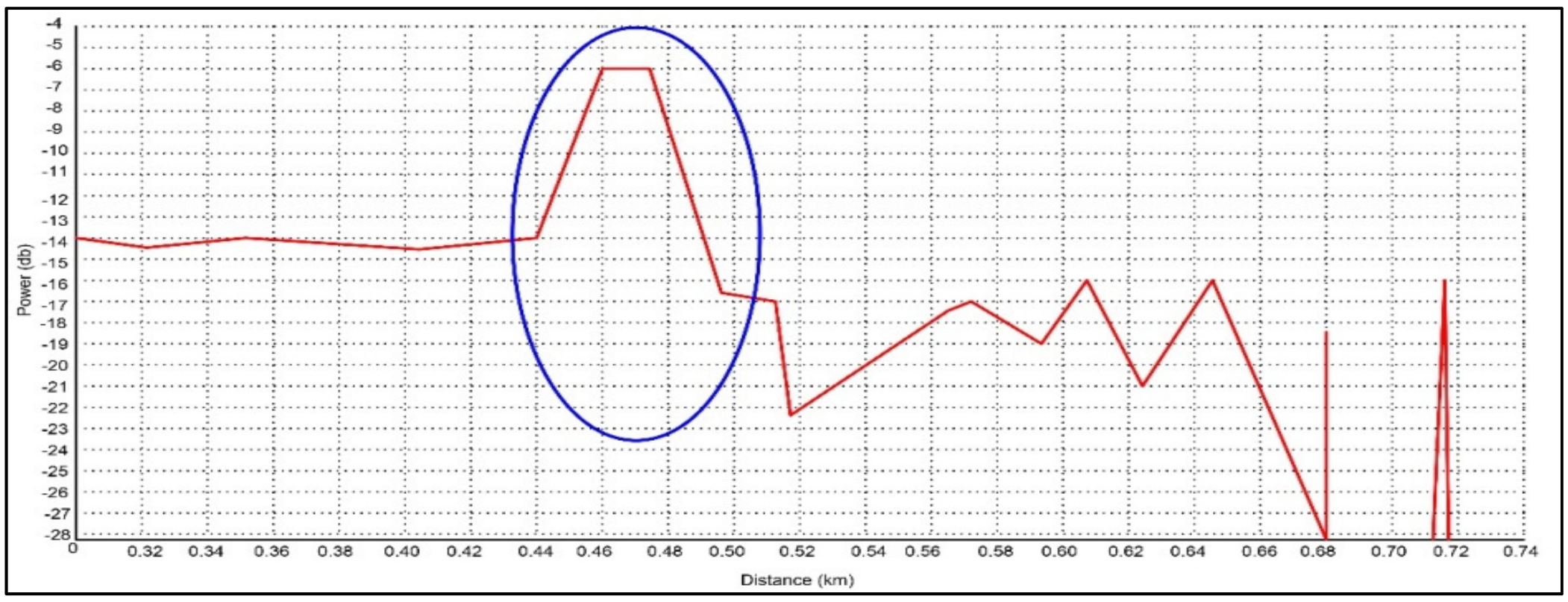


**Figure 11.** Event is Fiber break (standard SMF G.655 at 1550 nm was used and FPMT test signal was transmitted by the OSC board).

(d) Large insertion loss on the connector: It was already known that there was a large insertion loss on the connector created on a real applied system at a distance of 1.2 Km and the practical results using the FPMT technique are shown in Figure 12. The reflection peak was about 1.2 km and the reflection value was −3 dB which indicated a specific pattern expressing the large insertion loss on the connector failure. The test was performed on the standard type of SMF-G.652.

| | |
|---|---|
| Test signal transmitted by | OSC board |
| SMF Standard type is | SMF-G.652 |
| Wavelength used | 1550 nm |
| Failure detected by FPMT | Large insertion loss on the connector with a specific pattern drawn |
| FPMT fault location distance | Approx. 1.21 Km |
| Reflection value | Approx. −3 dB |

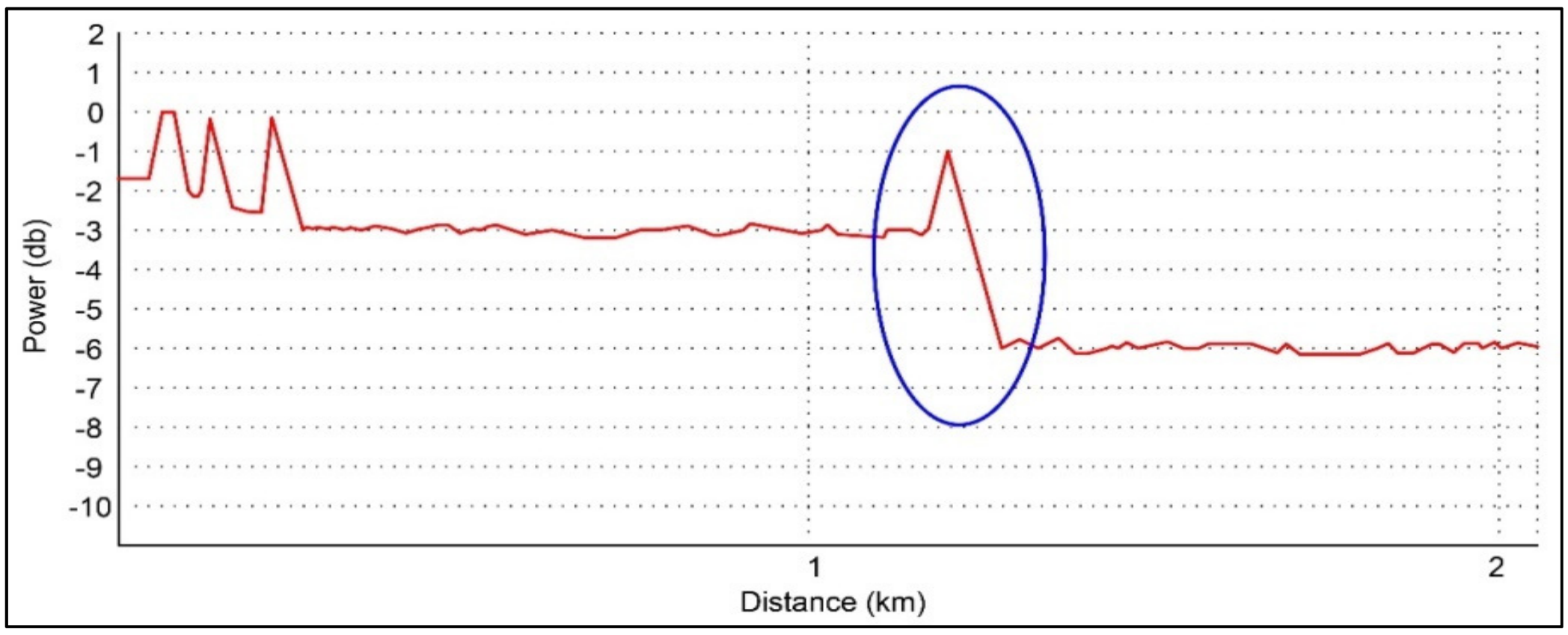


**Figure 12.** Event is fiber large insertion loss on the connector the (standard SMF G.652 at 1550 nm was used and FPMT test signal was transmitted by the OSC board).

(e) Interconnection (mismatch) between fiber cables: It was already known that there was a fiber cable mismatch created on a real applied system at a distance of 0.24 Km and the practical results using the FPMT technique are shown in Figure 13. The reflection peak was about 0.24 km and the reflection value was −2.4 dB which indicated a specific pattern expressing the interconnection (mismatch) between fiber cables failure. The test was performed on the standard type of SMF-G.652 and SMF-G.653.

| | |
|---|---|
| Test signal transmitted by | OSC board |
| SMF Standard type is | SMF-G.652 and SMF-G.653 |
| Wavelength used | 1550 nm |
| Failure detected by FPMT | Interconnection between fiber cables with a specific pattern drawn |
| FPMT fault location distance | Approx. 0.24 Km |
| Reflection value | Approx. −2.4 dB |

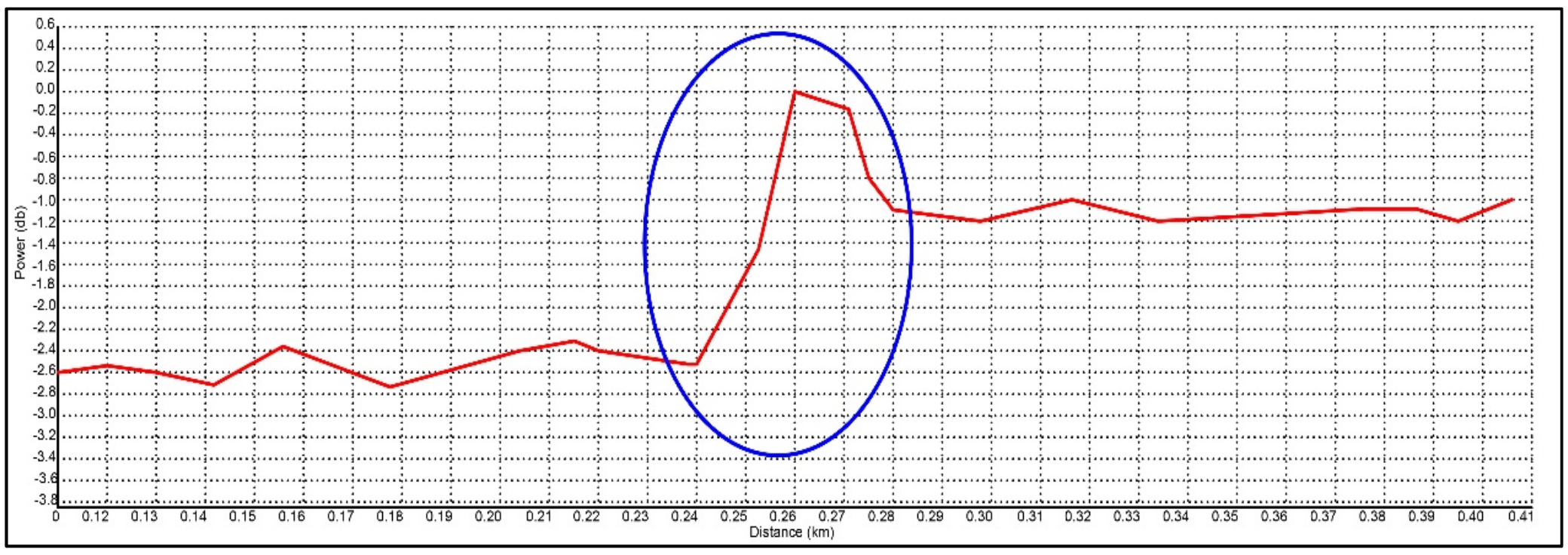


**Figure 13.** The fiber interconnection event between fiber cables G652 and G653 and the FPMT test signals transmitted by the OSC board.

### *5.2. Failurs Detection*

According to the practical results and feedback analysis of the reflected test signals, we found that the applied FPMT detected (real-time and remote) fiber failures over the fiber lines based on the pattern of the reflected test signal for each fault. The authors found a specific failure pattern for each event by repeating the experiment and re-collecting the practical results. Hence, it was noticed that each failure during fiber line detection had a certain pattern. Figure 14 depicts all these patterns of the reflected test signal that were observed during fiber line detection.

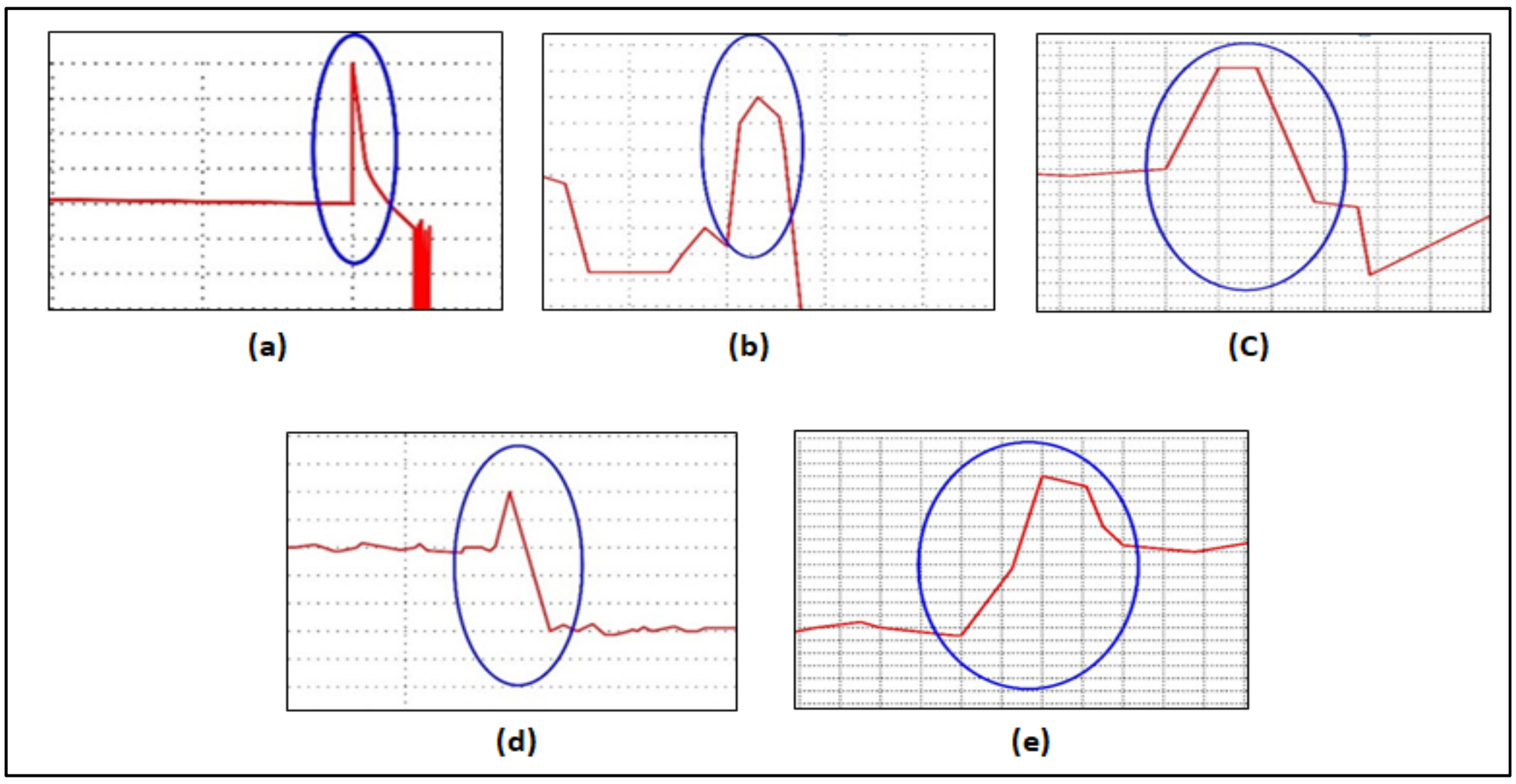


**Figure 14.** Pattern of the reflected test signals during fiber line detection.

### 5.3. Fault Location and Tolerance

The difference in the fault localization measured between the FPMT results compared to the actual faulty localization was as follows: the tolerance at (event $a_1$) was around 0.02 km which was the difference between both the practical measurement of proposed FPMT and actual fault locations were 5.22 km and 5.20 km, respectively. The tolerance at (event d) was around 0.01 km which was the difference between both practical measurement of proposed FPMT and the actual fault locations were 1.21 km and 1.21 km, respectively.

The difference obtained in FPMT measurements and the actual results had a very small tolerance at events ($a_1$, d) and had no tolerance for other events. Figure 15 depicts the distance of faults measured with the FPMT technique and the actual distance created on the system at different fiber events.

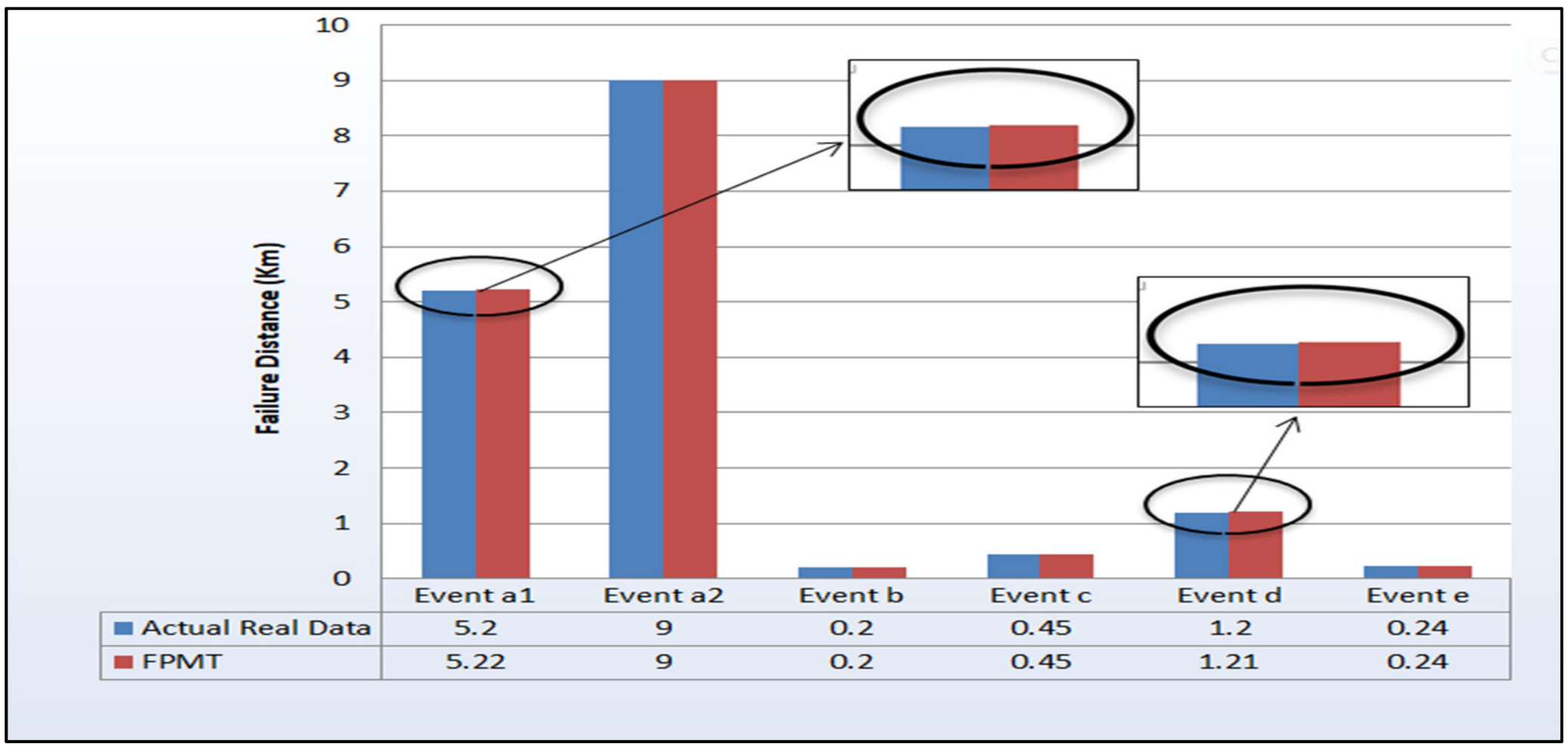


| | Event a1 | Event a2 | Event b | Event c | Event d | Event e |
|---|---|---|---|---|---|---|
| Actual Real Data | 5.2 | 9 | 0.2 | 0.45 | 1.2 | 0.24 |
| FPMT | 5.22 | 9 | 0.2 | 0.45 | 1.21 | 0.24 |

**Figure 15.** Real fault locations and fault locations detected by FPMT.

Deviation of distance fiber fault (km) = distance measured by FPMT technique—the fault location created on system.

According to the practical results shown in Figure 16, the deviation to the real fiber fault location using the FPMT technique was 10 to 20 m.

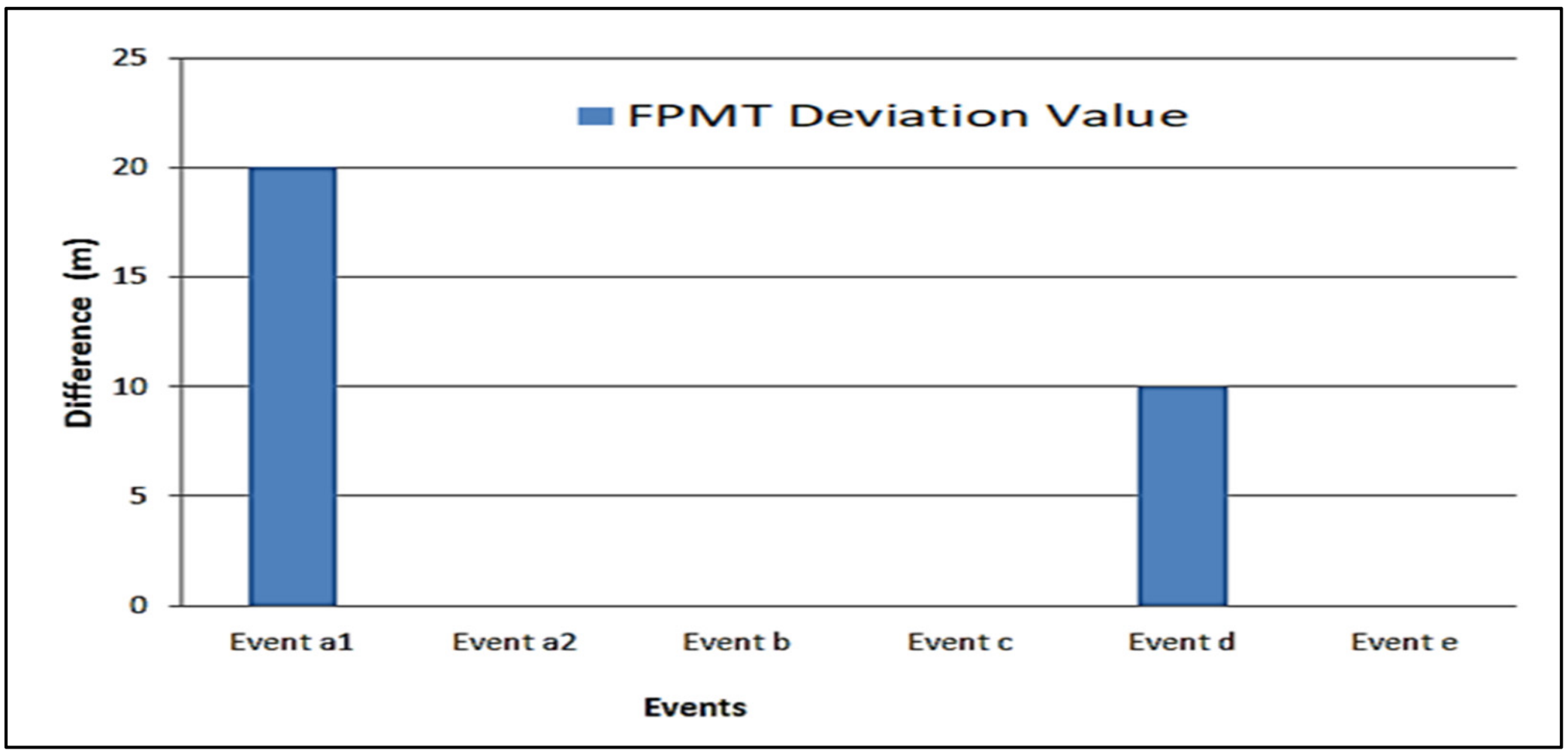


**Figure 16.** Difference measurement between Actual and FPMT.

### 5.4. Measurement Accuracy

The faults were localized by using the proposed FPMT technique with full accuracy of the event ($a_2$, b, c, e) up to 100% and with a high accuracy of the event ($a_1$, d) between 99.6 and 99.2, respectively.

The measurement accuracy for each event is depicted in Figure 17 with an average measurement accuracy of up to 99.8%.

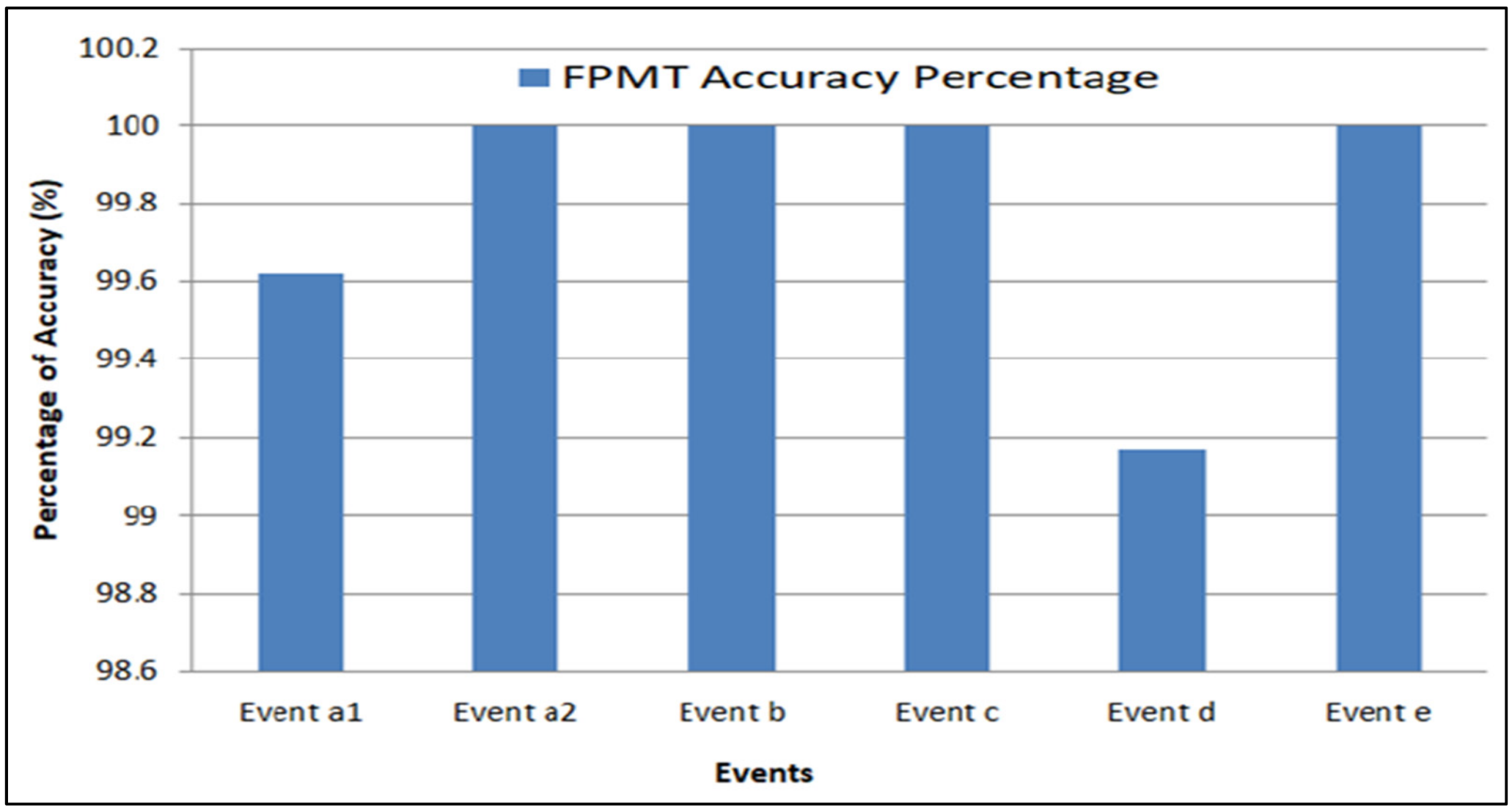


**Figure 17.** FPMT measurement accuracy.

### 5.5. Technique Comparison

There are two methods to detect fiber line faults: the traditional method of OTDR or the undeveloped online techniques briefly discussed in Section 1.1.

The practical results showed the flexibility and easiness of the FPMT technique to detect failures remotely and in real time without interrupting the traffic flow. Therefore, the FPMT technique is comparatively better than the OTDR method in terms of detecting fiber failures remotely and in real time in a way that minimizes network downtime and restores the system rapidly, unlike the costly OTDR technique which is manually operated to detect fiber failures which leads to more effort and time.

Table 7 shows the proposed FPMT technique compared to the traditional OTDR technique.

**Table 7.** Comparison between proposed FPMT technique and traditional OTDR technique.

| | **FPMT Technique** | **OTDR Technique** |
|---|---|---|
| Data traffic flow | Not interrupting data flow | Interrupting data flow |
| Reliable detection | Online | Offline |
| Fiber failure access speed | Remotely | Manually |
| Operation and maintenance expenses | Low cost | High cost, requires more time and effort |

The practical results also showed that the proposed FPMT techniques were distinguished from the online FBMS technique at the following measurements: both FPMT and FBMS at the system had power budget insertion losses of 0.4 dB and 1.7 dB, respectively, accuracies of fault localization measurement were 99.8% and 69.85%, respectively, tolerance deviations of 0.02 km and 0.285 km, respectively, and maximum distances to detect fiber line faults of 150 km and 4.49 km, respectively.

Table 8 shows the proposed FPMT technique compared to the FBMS online technique.

**Table 8.** Comparison between the proposed FPMT technique and the FBMS online technique.

| Type | FPMT Technique | FBMS Technique |
|---|---|---|
| Max. distance can be detected | 150 Km | 4.49 Km |
| Average of fiber fault location measurement accuracy (%) | 99.8% | 96.85% |
| Deviation | ±0.02 Km | ±0.285 Km |
| Insertion loss | 0.4 dB | 1.7 dB |
| Types of fiber failure detected | 5 different types | fiber break only |

## 6. Conclusions

This paper reviewed the new technique for monitoring fiber performance. The authors proposed and evaluated a new FPMT technique that was designed to detect, locate, and estimate optical hard failures through the passive optical network and single mode fibers. This technique worked in real time remotely without interrupting the traffic flow and with low effort and costs. The proposed FPMT technique featured an improved system power budget with a minimal insertion loss of 0.4 dB, detected fiber faults with an average accuracy of fault localization measurement up to 99.8% with a small deviation of 10–20 m, and a maximum distance to detect fiber line faults of up to 150 km. FPMT detected multiple types of fiber faults such as fiber breaks, fiber end face contamination, fiber end face burning, and large insertion losses on the connector and interconnection between two different fiber cables. The proposed technique demonstrated a better performance in the following parameters: improving the system power budget with a low insertion loss, supporting a higher range of distance testing and performance monitoring, a high measurement accuracy, a small deviation value, and a multiple types of fiber faults detected.

*Future Work*

In future work, the FPMT technique should be reapplied to detect all optical network failure patterns. A new alter-native board at the DWDM system could be integrated with the FPMT technique to support a higher range of distance testing. The concept of this method in future research could be used with the machine learning algorithm to predict failure pattern shapes without the need for quantitative parameters.

**Author Contributions:** Conceptualization, methodology, A.A.I. and M.M.F.; investigation, resources, data analysis, A.A.I.; data curation, A.A.H.; writing—original draft preparation, A.A.I. and M.M.F.; results tabulate and graphic presentation, M.M.F., A.A.I. and A.A.H.; writing—review and editing, A.A.I. and A.A.H.; visualization, A.A.I.; supervision, M.M.F. and A.A.H.; All authors have read and agreed to the published version of the manuscript.

**Funding:** This research received no external funding.

**Institutional Review Board Statement:** Not applicable.

**Informed Consent Statement:** Not applicable.

**Data Availability Statement:** The study did not report any data.

**Acknowledgments:** The authors are grateful to Mohamed Saed at the Huawei Egypt research department for providing technical support for this work.

**Conflicts of Interest:** The authors declare no conflict of interest.

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
