# Peer review of "A Design Fiber Performance Monitoring Tool (FPMT) for Online Remote Fiber Line Performance Detection"

_electronics, doi:10.3390/electronics11213627_

Round 1

Reviewer 1 Report

Fiber faults events detection and monitoring in optical communication network systems is important. In this article, the authors proposed a technique for online remote fiber line detection, however, the technique is similar to OTDR (the principle is almost the same as OTDR), the popular technique for fiber detection. The test signals are modulated in a laser with frequency different to that of traffic signal. Therefore, there is no interference with the traffic signal transmitting on the fiber, and the technique is an online detection.

Though the proposed method seems not very novel and the principle is not impressed, this work could become better (and more significance), provided detailed experimental investigation are conducted. I’m confused about the results shown in Section 5. Are they experimental results or numerical results? If experiments were conducted, I suggest the authors carry out more investigations. As shown in Fig. 14, different patterns of the reflected signal are obtained, corresponding to different fiber faults events. Why the patterns are different? Is there any quantitative parameter, instead of the qualitative figures shown in Fig. 14, distinguishing different type of fiber faults?

By the way, I suggest the authors describing the principle of the proposed technique in the abstract and the introduction part.

Author Response

The authors will be taken the most observations and will be add in the final submit manuscript.

Reviewer 2 Report

The study presented in the manuscript is interesting and brings novelty into the investigated domain.

Although the scientific proof is well constructed by the authors, the English language must be definitely improved in the entire manuscript. Here are some of the English errors I noticed:

Line 126

Include Indentation for the first line (Tab)

Line 132

Replace in the sentence

The interleaver board (ITL) it’s used to multiplex/combine the even and odd frequency

With

The interleaver board (ITL) is used to multiplex/combine the even and odd frequency

Lines 135 - 137

Replace in the sentence

The optical amplifier (OA) it’s used to boost the signal power

With

The optical amplifier (OA) is used to boost the signal power

Line 135

Replace the sentence

The optical amplifier (OA) it’s used to boost the signal power and al- 135 lowing it to travel over a long distance can be used the Raman amplifier or erbium doped 136 fiber (EDFA) amplifier.

with

The optical amplifier (OA) is used to boost the signal power and allow it to travel over a long distance. The Raman amplifier or erbium doped fiber (EDFA) amplifier can be used.

Line 137

Replace in the sentence

The dispersion compensation module (DCM) it’s used to compensate the dispersion of the signal

With

The dispersion compensation module (DCM) is used to compensate the dispersion of the signal

Line 139

Replace in the sentence

supervisory channel board (OSC) it’s used

with

supervisory channel board (OSC) is used

Line 144

Replace in the sentence

The fiber interface board (FIU) it’s used to combine/multiplex

With

The fiber interface board (FIU) is used to combine/multiplex

One figure (1 or 2 is enough)

Line 188 Does not have sense. As follows... and there is only one item!

Line 190 The laser diode (LD) it’s received a digital stream from RSG…

Replace with The laser diode (LD) receives a digital stream from RSG and …

Line 192

Replace The hybrid filter (HF) it’s distinguishes

 with The hybrid filter (HF) distinguishes

Line 194

Replace The photo detector (PD) it’s receives

With The photodetector (PD) receives

Line 255

network via using three ways as follows;

network via using three ways as follows; .

Line 267

to avoid interference with control and traffic signals, this way can be considered merging between  first way and second way and more safe for any interference.

Replace with

to avoid interference with control and traffic signals. This method can be considered a mixture between the first method and second method and is safer for any type of interference.

Line 275

has shown in Figure 5

Replace with

is shown in Figure 5

Only fig 6 should remain in manuscript. Reach a clear view of fig 6 as in fig 5.

Lines 358 and 359

I believe you meant per kilometer, not per kilometers

Line 369

as follows;

Usually after as follows this punctuation : appears not ;

Lines 389 and 390

The practical results collected by Huawei network cloud engine (NCE) server’s at 389 optical transmission equipment according to analyze the reflected test signals.

I believe you meant:

The practical results collected by Huawei network cloud engine (NCE) server at the optical transmission equipment are according to the reflected test signals analyzed.

Line 391

The system implemented through access remotely Huawei labs infrastructure nodes.

I believe you meant:

The system is implemented through access remotely Huawei labs infrastructure nodes.

Lines 397 398

The specification of the implemented system for monitoring and analyzing the fiber 397 line and collecting the information has shown in Table1 and Table 5.

I believe you meant to say:

The specifications of the implemented system for monitoring and analyzing the fiber line and collecting the information are shown in Table1 and Table 5.

Author Response

Finally; The authors will be taken the most observations and will be add in the final submit manuscript.

Reviewer 3 Report

The authors proposed a new method and technique for online detection of optical fibers defects, which is better, cheaper and more convenient than previous known methods. The idea of detection of the fiber defects is understandable and presented in logical way. Experimental results are presented in graphs and tables. There are some comments on the quality of the design of the paper, the authors should increase the quality of the drawings.

The comments are follows.

1.      Line 162, 178. Text is invisible.

2.      It seems better to do some vertical compression of Fig.3, Fig.7, and horizontal compression of subtitles (a), (b),…(e) on the Fig. 14.

3.      There are 17 figures in the text, and the style of figures (font style, font size, style of schemes, resolution of pics, horizontal and vertical stretching of letters) is different. It seems to me the  design of the figures and resolution of some (f.e. fig. 6, 14) may be improved.  

4.      The sign “dash” (-) should be replaced with the sign “minus” (f.e. at values -7dB, -34dB  etc.) .

5.      Scripts in formulae differ from the script of main text, commas are far from formulae. It is better to retype them.

6.      The tables under lines 418, 432,446, 460 do not have any titles.

7.      Lines 428, 441 , 456, 469, 484: some letters are in the bold style.

In general, the work is interesting to me and relevant. I wish the authors success in their further research.

Author Response

(The authors gave the same response as above.)

Round 2

Reviewer 1 Report

Though the principle of the proposed scheme is similar to OTDR, detailed experimental investigations were conducted and reported in the revised article.